# Navigating the Digital Neurolandscape: Analyzing the Social Perception of and Sentiments Regarding Neurological Disorders through Topic Modeling and Unsupervised Research Using Twitter

Javier Domingo-Espiñeira [1], Oscar Fraile-Martínez [1,2,*], Cielo Garcia-Montero [1,2], María Montero [1], Andrea Varaona [1], Francisco J. Lara-Abelenda [1,3], Miguel A. Ortega [1,2], Melchor Alvarez-Mon [1,2,4] and Miguel Angel Alvarez-Mon [1,2,5]

[1] Department of Medicine and Medical Specialities, Faculty of Medicine and Health Sciences, University of Alcala, 28801 Alcala de Henares, Spain; javier.domingo@iese.net (J.D.-E.); cielo.garcia@uah.es (C.G.-M.); maria.monterotorres@gmail.com (M.M.); varaonaandrea@gmail.com (A.V.); franciscolaraabelenda@gmail.com (F.J.L.-A.); miguelangel.ortega@uah.es (M.A.O.); melchor.alvarezdemon@uah.es (M.A.-M.); miguelangel.alvarezm@uah.es (M.A.A.-M.)

[2] Ramón y Cajal Institute of Sanitary Research (IRYCIS), 28034 Madrid, Spain

[3] Departamento Teoria de la Señal y Comunicaciones y Sistemas Telemáticos y Computación, Escuela Tecnica Superior de Ingenieria de Telecomunicación, Universidad Rey Juan Carlos, 28942 Fuenlabrada, Spain

[4] Service of Internal Medicine and Immune System Diseases-Rheumatology, University Hospital Príncipe de Asturias, (CIBEREHD), 28806 Alcala de Henares, Spain

[5] Department of Psychiatry and Mental Health, Hospital Universitario Infanta Leonor, 28031 Madrid, Spain

* Correspondence: oscarfra.7@hotmail.com

**Abstract:** Neurological disorders represent the primary cause of disability and the secondary cause of mortality globally. The incidence and prevalence of the most notable neurological disorders are growing rapidly. Considering their social and public perception by using different platforms like Twitter can have a huge impact on the patients, relatives, caregivers and professionals involved in the multidisciplinary management of neurological disorders. In this study, we collected and analyzed all tweets posted in English or Spanish, between 2007 and 2023, referring to headache disorders, dementia, epilepsy, multiple sclerosis, spinal cord injury or Parkinson's disease using a search engine that has access to 100% of the publicly available tweets. The aim of our work was to deepen our understanding of the public perception of neurological disorders by addressing three major objectives: (1) analyzing the number and temporal evolution of both English and Spanish tweets discussing the most notable neurological disorders (dementias, Parkinson's disease, multiple sclerosis, spinal cord injury, epilepsy and headache disorders); (2) determining the main thematic content of the Twitter posts and the interest they generated temporally by using topic modeling; and (3) analyzing the sentiments associated with the different topics that were previously collected. Our results show that dementias were, by far, the most common neurological disorders whose treatment was discussed on Twitter, and that the most discussed topics in the tweets included the impact of neurological diseases on patients and relatives, claims to increase public awareness, social support and research, activities to ameliorate disease development and existent/potential treatments or approaches to neurological disorders, with a significant number of the tweets showing negative emotions like fear, anger and sadness, and some also demonstrating positive emotions like joy. Thus, our study shows that not only is Twitter an important and active platform implicated in the dissemination and normalization of neurological disorders, but also that the number of tweets discussing these different entities is quite inequitable, and that a greater intervention and more accurate dissemination of information by different figures and professionals on social media could help to convey a better understanding of the current state, and to project the future state, of neurological diseases for the general public.

**Keywords:** Twitter; neurological disorders; dementias; topic modeling; sentiment analysis

## 1. Introduction

Neurological disorders encompass a group of conditions that represent the leading cause of disability and the second leading cause of death worldwide [1]. Alzheimer's and other dementias, Parkinson's disease, multiple sclerosis, spinal cord injury, epilepsy and headache disorders are the central neurological disorders that are continuously growing in developed, low-income and middle-income countries, thus entailing an emerging challenge for healthcare systems and evidencing a need for additional research, as well as a broader understanding of these disorders' impact from different perspectives [2]. Considering the social perceptions and public opinions related to neurological disorders could have a great impact on patients and their relatives and caregivers, since how diseases are seen in society or the perceived support can decisively influence well-being, quality of life and the ability to cope with the disease, as well as raising awareness and transmitting the roles of different professionals involved in the multidisciplinary management of these conditions [3–6].

A growing body of evidence indicates that social networks can help us to understand social opinions about various topics from a closer perspective, reporting some notable benefits when compared to other sources, including increased social interactions; peer, social and emotional support; and broader accessibility to shared and tailored health-related information [7]. Additionally, Twitter offers some unique benefits that have been documented in earlier studies. It is a useful tool for fostering a sense of community, increasing awareness of various topics, providing a safe space for expression and maybe serving as a channel of communication between patients, families and healthcare professionals [8–10]. The use of Twitter in the research field of neuropsychiatric diseases has also been studied in the available literature [11,12]. Twitter has brought multiple benefits, such as allowing neurologists and other professionals to educate and share information and research related to neurological diseases with a broader global audience and to improve patient care and support [13]. However, as it is easily accessible to the general population, misinformation and the promotion of vested interests can be too easily and widely disseminated, as well as the occurrence of inappropriate and unprofessional conduct, due to the lack of regulations for social media use established by institutional, national or ethical/legal guidelines. This type of approach has been replicated for specific types of neurological diseases, such as Alzheimer's and other dementias [14], migraines [15], epilepsy [16] and less common neurological disorders [17,18]. Nonetheless, there is little research on the information on Twitter about collectively treating the most notable neurological disorders or related sentiments, as well as the nature of the conversations about and social perceptions of neurological disorders in general. In this sense, the aim of the present study is to (1) compare the numbers of tweets and their temporal evolution regarding the most notable neurological disorders (dementias, Parkinson's disease, multiple sclerosis, spinal cord injury, epilepsy and headache disorders); (2) determine the main thematic content of the Twitter posts and the interest they generated; and (3) analyze the feelings associated with these different topics. To represent different regions and cultures, both English and Spanish tweets are included in our study.

## 2. Materials and Methods

### 2.1. Search Strategy and Data Collection on Twitter

We collected and analyzed all the tweets posted in English or Spanish, between 2007 and 2023, referring to headache disorders, dementia, epilepsy, multiple sclerosis, spinal cord injury or Parkinson's disease using a search engine that has access to 100% of the publicly available tweets. The inclusion criteria for the tweets in our study were as follows: (1) including the aforementioned keywords; (2) public accessibility; (3) being written in

either Spanish or English; and (4) publication within the temporal span ranging from 2007 to 2023.

## 2.2. Tweet Analysis with Natural Language Processing and Topic Modeling

This study adopted an unsupervised learning approach using LDA for topic modeling. Prior to the analysis of the data with the LDA model, an extensive data preprocessing procedure was implemented. This preprocessing encompassed language classification, segregating the Spanish tweets from the others and subsequently translating the remaining tweets into English using the Google Translator application. Following this, the data cleansing activities included the removal of duplicated words, stop words and extraneous textual elements, such as hashtags, emojis and non-standard characters. To determine the optimal number of topics for modeling, a cluster validity index (CVI) was employed. CVIs are measures used in unsupervised learning to evaluate the effectiveness of clustering solutions by assessing the quality of the data points [19]. The selected CVI was the silhouette coefficient, which ranges between $-1$ and 1, with higher values indicating a better clustering performance. The silhouette coefficient was selected for its ability to assess both inter-cluster and intra-cluster distances. Subsequently, LDA was applied to the six neurological diseases analyzed (Parkinson's, dementias, headaches, epilepsy, multiple sclerosis and spinal cord injury) for both the Spanish and English datasets, resulting in a total of 12 topic modeling applications. Lastly, a sentiment analysis was conducted using models from Hugging Face's machine learning platform, specifically "Emotion English DistilRoBERTa-base" [20] for the English dataset and "Beto emotion analysis" for the Spanish dataset, based on the BETO Base model [21,22]. These models facilitated the categorization of the tweets on each topic into Ekman's six fundamental emotions—anger, disgust, fear, joy, sadness and surprise—with the addition of the neutral emotion [23].

## 2.3. Ethical Considerations

This study was initially reviewed by the University of Alcala Research Ethics Committee and was determined to be research that did not involve patients. It is compliant with the research ethics principles of the Declaration of Helsinki (seventh revision, 2013).

## 3. Results

### 3.1. Dementia Was, by Far, the Most Common Neurological Disorder Commented on in the English and Spanish Tweets, but Other Neurological Disorders Received Broader Attention over Time

Firstly, we aimed to evaluate the numbers of English and Spanish tweets discussing the main neurological disorders over time. We observed that, by far, dementia was the most commented on disease on Twitter, accounting for up to 95,208 tweets in Spanish and 167,943 tweets in English. Epilepsy and Parkinson's disease completed the rankings in both languages, whereas spinal cord injury was the neurological disorder with the smallest number of tweets (Figure 1).

Regarding its temporal evolution, we observed that mentions of dementia reached two peaks in the Spanish tweets: one in 2013 and another in 2020 (Figure 2). It was also observed that mentions of Parkinson's disease also reached a significant peak in 2022. For the English tweets, there was only one peak, also in 2020. Epilepsy and Parkinson's disease displayed increased interest in both the Spanish and English tweets, while the latter had a peak in 2022 (Figure 2).

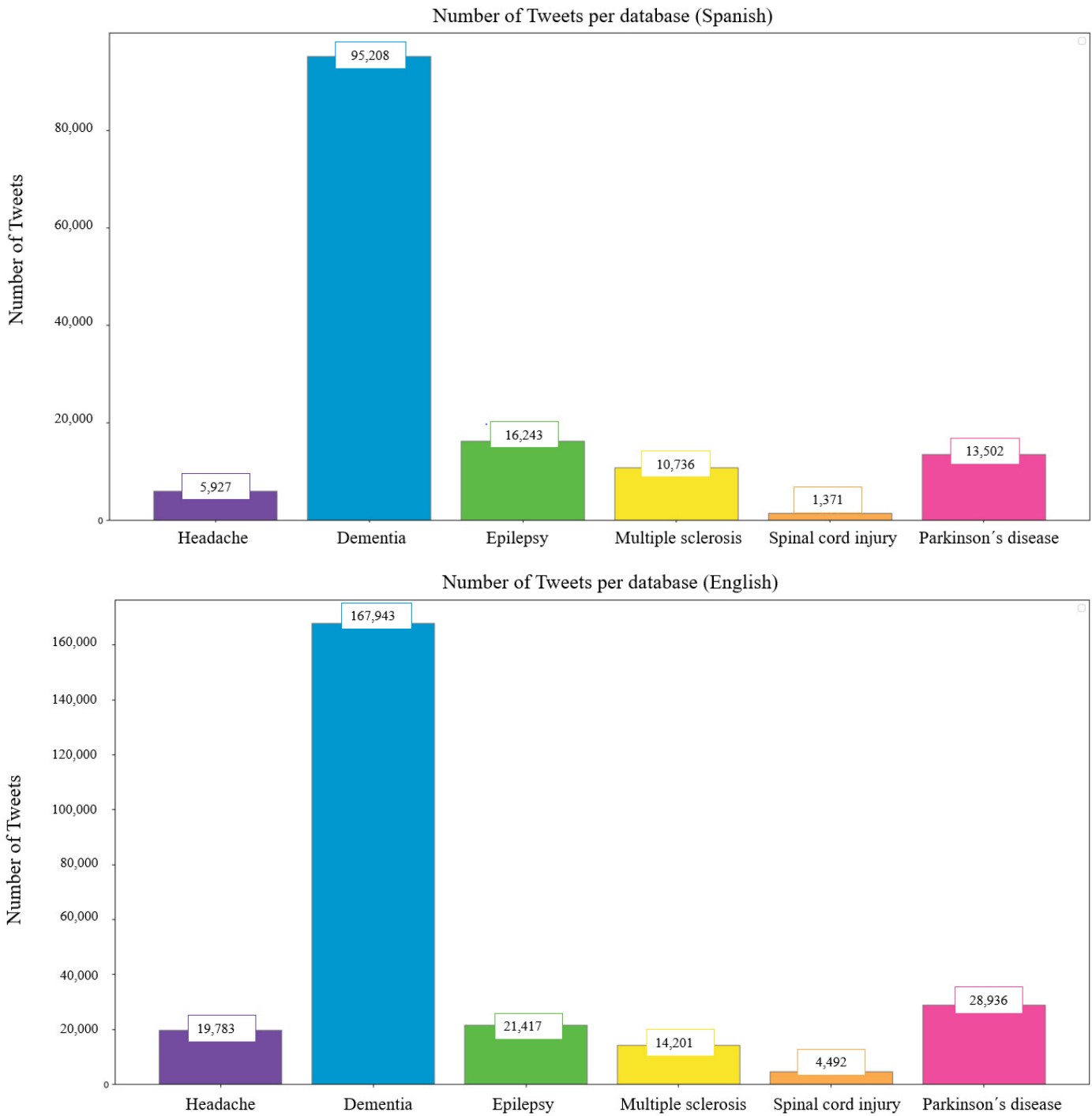

**Figure 1.** Number of tweets per database in English and Spanish.

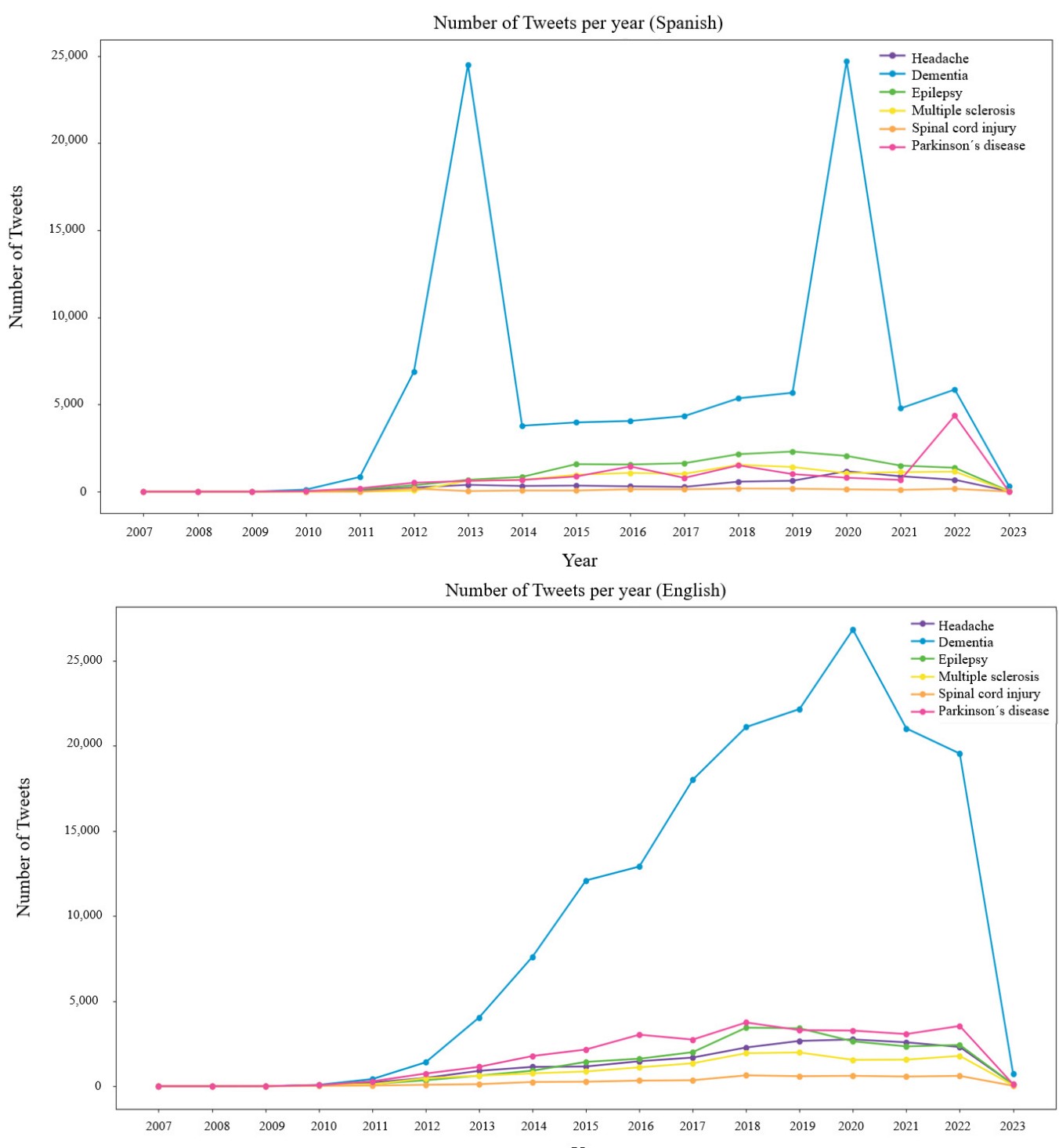

**Figure 2.** Number of tweets per year in English and Spanish.

### 3.2. Topic Modeling Analysis

3.2.1. Main Topics and Temporal Evolution of Associated Tweets

After the performance of the topic modeling analysis of the studied tweets, the most frequent themes were identified and ordered from the greatest to least number of tweets received in both languages (Figure 3). In Spanish (Figure 3A), the most frequent topics included the personal experiences of relatives of patients with neurological diseases (Topic 1), closely followed by the need for/use of certain medications (sinemet, mirapex, stalevo, etc.) for the treatment of the neurological diseases treated in this analysis—mainly Parkin-

son's disease (Topic 4)—and the impact and fight against neurological diseases (Topic 0). Topic 3 discussed the dissemination of information through the media, events and famous people to raise awareness and show the impact of neurological diseases, whereas the least commented on topic by far was related to patient and family associations aiming to increase social awareness about neurological diseases (Topic 2). Meanwhile, in English (Figure 3B), the most common topic was family associations that aim to support families, raise awareness and facilitate the research of neurological diseases (Topic 1), followed by the exploration of new treatments (marijuana, exercise, etc.), the unraveling of the potential causes of neurological disorders (Topic 3), the research on neurological disorders (Topic 2) and charity fundraising organizations for the study and research of neurological disorders (Topic 0). The least commonly commented on topic involved famous people with or known cases of suffering from neurological disorders (Topic 4).

Then, we evaluated the temporal evolution of each topic observed in our study. In the Spanish tweets, we observed that since 2010, Topic 1 has presented with almost continuous growth without any significant peak, opposite to Topic 2, in which two clear peaks can be observed, one in 2012 and another in 2020. Topic 0 showed a similar pattern to Topic 1 and, despite being one of the less abundant topics, Topic 3 had the highest peak among the Spanish tweets in 2013, as observed in Figure 4A. For the English tweets, we can report that Topic 1 reached its highest peak in 2020, and is the one that has accumulated the most tweets since 2019. Topic 3 was the most common topic between 2013 and 2018, when it reached its maximum peak, whereas Topic 2 has always been ranked as the second or third most common topic, except in 2020, when Topic 0 had its maximum peak. As can be observed, Topic 4 has always placed in the last position (Figure 4B).

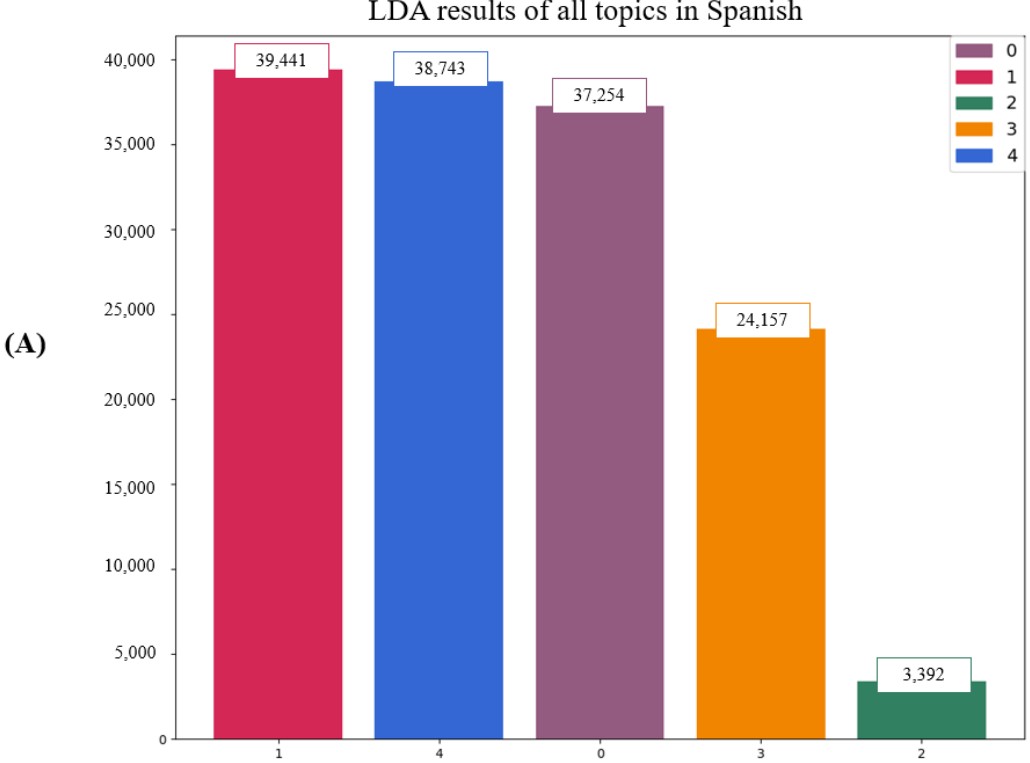

**Figure 3.** *Cont.*

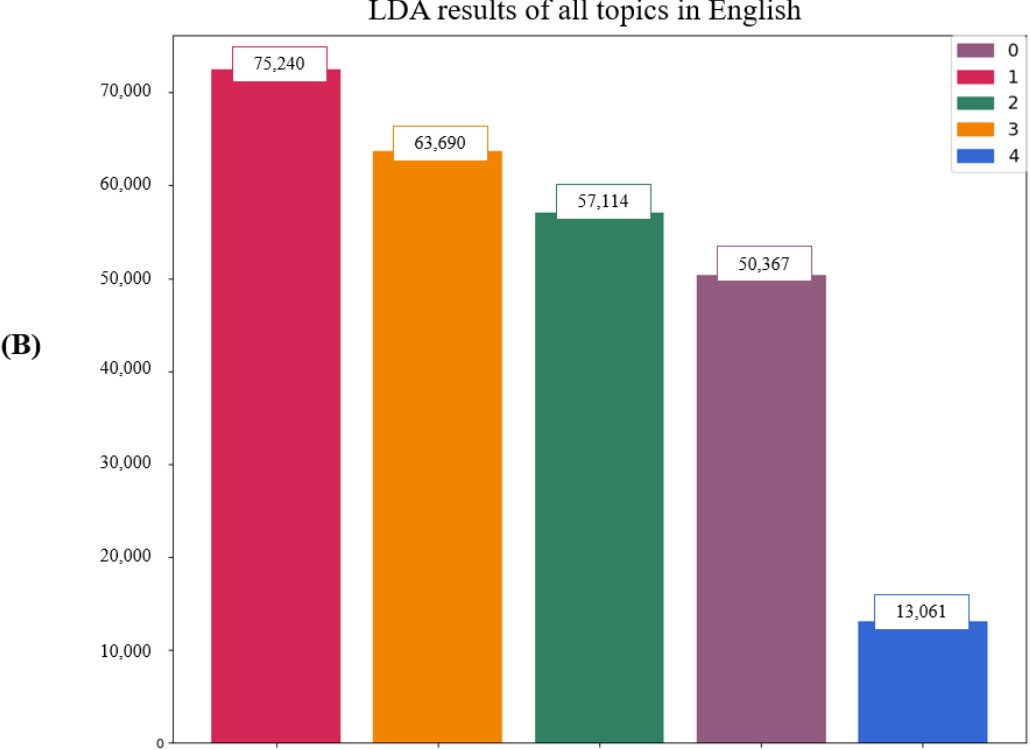

**Figure 3.** Main topics described in our study. Legend: (**A**) Spanish tweets. Topic 1 (pink): personal experiences of relatives of patients with neurological diseases; Topic 4 (blue): medications used for neurological disorders; Topic 0 (purple): impact of and fight against neurological diseases; Topic 3 (orange): dissemination of information about neurological disorders by media, events and famous people; Topic 4 (green): patient and family associations. (**B**) English tweets. Topic 1 (pink): support for families and raising awareness and encouraging research of neurological diseases; Topic 3 (orange): new treatments for and pathophysiologies of neurological disorders; Topic 2 (green): research on neurological disorders; Topic 0 (purple): charity fundraising organizations for the study and research of neurological disorders; Topic 4: famous people with or known cases of neurological disorders.

3.2.2. Sentiment Analysis

A sentiment analysis was performed for each relevant theme identified. The feelings studied were fear, sadness, joy, anger, surprise and disgust. The evolution of these feelings were evaluated in relation to the most relevant topics selected over the years.

In the sentiment analysis of the tweets that addressed the most important topics found in the topic modeling, various associated emotions were observed and ordered in their importance (Figure 5). In the Spanish tweets (Figure 5A), patient and family associations aiming to increase social awareness about neurological diseases (Topic 2) mainly provoked emotions of anger and sadness. In parallel, the need/use of certain medications (sinemet, mirapex, stalevo, etc.) for the treatment of neurological diseases was predominantly associated with feelings of joy. In English (Figure 5B), family associations that aimed to support families and raise awareness and facilitate the research of neurological diseases (Topic 1) mainly generated feelings of sadness and fear among users, although anger, surprise and joy were also detected in our study. The exploration of new treatments mainly aroused fear and sadness but also joy (Topic 3). Charity fundraising organizations for the study and research of neurological disorders (Topic 0) and the exploration of neurological disorders with AI (Topic 2) also generated feelings of fear, whereas famous cases of neurological disorders (Topic 4) were more related to feelings of sadness.

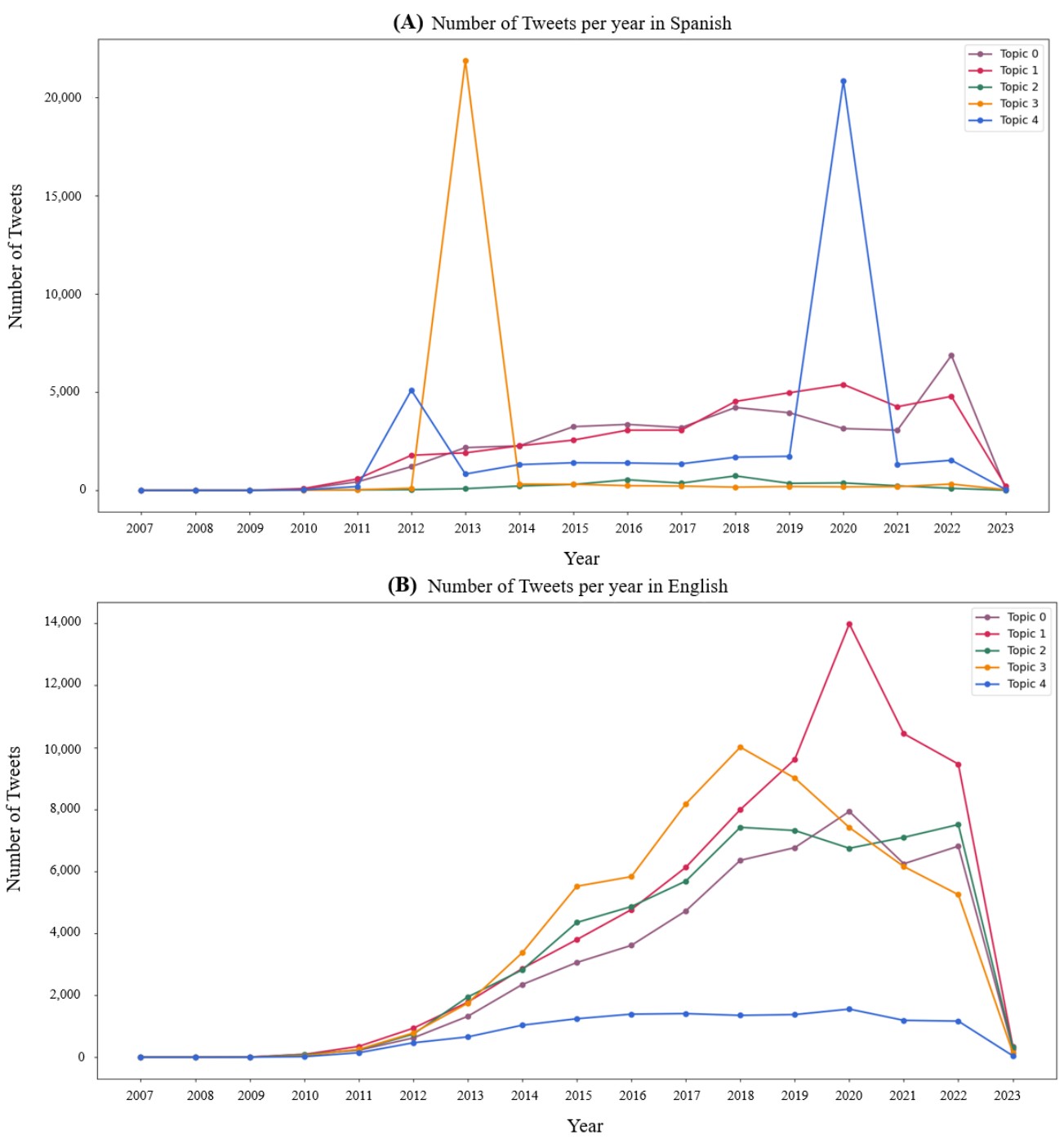

**Figure 4.** Temporal evolution of the main topics described in our study. (**A**) Spanish tweets. Topic 1 (pink): personal experiences of relatives of patients with neurological diseases; Topic 4 (blue): medications used for neurological disorders; Topic 0 (purple): impact of and fight against neurological diseases; Topic 3 (orange): dissemination of information about neurological disorders by media, events and famous people; Topic 4 (green): patient and family associations. (**B**) English tweets. Topic 1 (pink): supporting families and raising awareness and encouraging research of neurological diseases; Topic 3 (orange): new treatments for and pathophysiologies of neurological disorders; Topic 2 (green): research on neurological disorders; Topic 0 (purple): charity fundraising organizations for the study and research of neurological disorders; Topic 4: famous people with or known cases of neurological disorders.

**(A)** Topic comparison in Spanish database

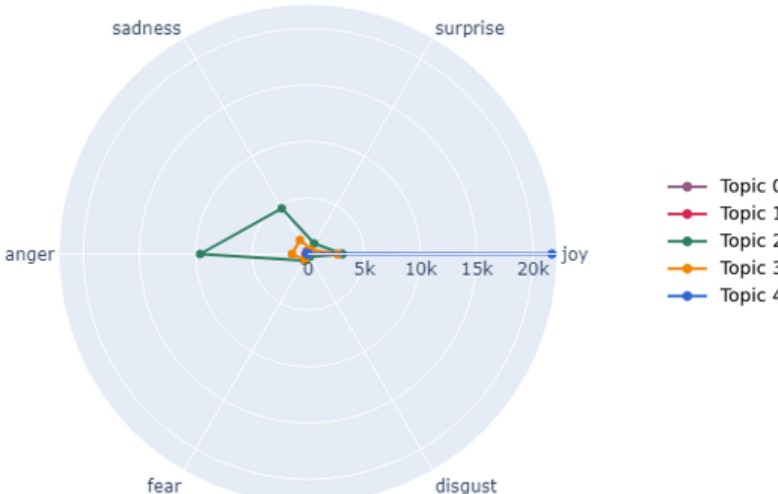

**(B)** Topic comparison in English database

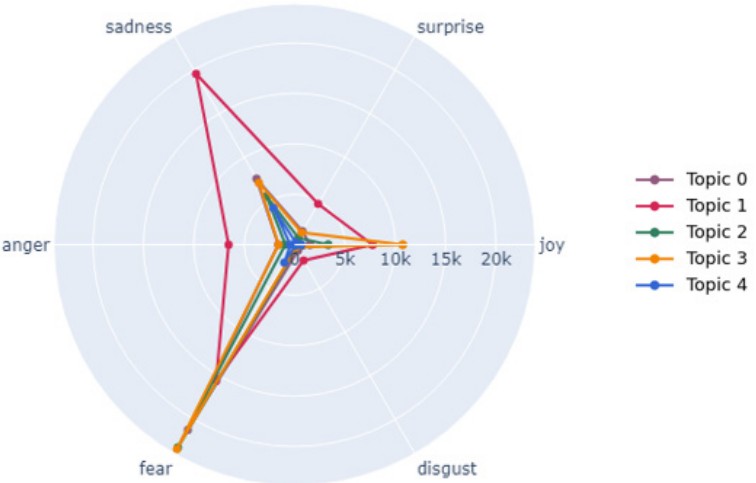

**Figure 5.** Sentiment analysis of the main topics. (**A**) Spanish tweets. Topic 1 (pink): personal experiences of relatives of patients with neurological diseases; Topic 4 (blue): medications used for neurological disorders; Topic 0 (purple): impact of and fight against neurological diseases; Topic 3 (orange): dissemination of information about neurological disorders by media, events and famous people; Topic 4 (green): patient and family associations. (**B**) English tweets. Topic 1 (pink): support for families and raising awareness and encouraging research of neurological diseases; Topic 3 (orange): new treatments for and pathophysiologies of neurological disorders; Topic 2 (green): research on neurological disorders; Topic 0 (purple): charity fundraising organizations for the study and research of neurological disorders; Topic 4: famous people with or known cases of neurological disorders.

### 4. Discussion

The incidence and prevalence of neurological disorders are growing at a worrying rate, having a huge impact on disability-adjusted life years (DALYs) and mortality worldwide [24]. Twitter and social media have the potential to influence health policies and raise social awareness about different concerns. In the present work, we have observed that most tweets discussing neurological disorders are focused on dementias, like Alzheimer's disease, followed by epilepsy, Parkinson's disease and migraines. In parallel, we have ob-

served that the impact of neurological diseases on patients and relatives, claims to increase public awareness, social support and research, activities to ameliorate disease development and existent/potential treatments or approaches for neurological disorders were the main topics discussed in tweets. Additionally, we have been able to identify the sentiments associated with these contents, reporting that fear, anger and sadness, but also positive feelings like joy, can be observed.

First, dementias were, by far, the most commented on neurological disorder category in our collected tweets. According to the existing literature, stroke, dementias and headache disorders (especially migraines) are the three most burdensome neurological disorder groups in developed countries like the United States [25]. Alzheimer's disease and other dementias represent the second group of neurological disorders, with the highest associated DALYs associated, just after stroke [25]. Currently, the medical management of dementia has an associated cost of USD 600 billion globally, with 47 million people in the world with dementias, although this number is expected to increase almost three times, up to 131 million, by 2050 [26]. In this sense, the overall impact of dementias is undeniable, and Twitter might reflect these facts. Previous works conducted on the Twittersphere have found that Alzheimer's disease and related dementias are broadly treated on this social network [27–29]. However, what was surprising for us was the great difference in the number of tweets discussing dementia in comparison to the remaining neurological diseases included in our study, which also have a significant impact on society. In particular, for the case of headache disorders, epidemiological data have ranked this entity (specifically migraines) as the second major cause of disability after back pain with respect to years of life lived with disability, accounting for an annual 3% of all emergency visits [30]. Thus, the great difference between the number of tweets about dementias and migraines is surprising, as prior works conducted on Twitter have claimed the relevance of this platform in regard to headache disorder discussions [31]. Similar conclusions can be drawn when evaluating the growing prevalence, incidence and burden of other neurological disorders, like Parkinson's disease, multiple sclerosis, epilepsy and spinal cord injury [24]. In the case of the latter, the low representation of this disease in tweets is especially striking, particularly due to the numerous psychosocial consequences for patients, relatives and healthcare professionals [32]. Thus, our results show great inequality on Twitter with respect to the consideration of different neurological diseases, with dementias being widely discussed and other entities, such as spinal cord injury, conspicuously ignored. A greater presence of users, professionals or platforms aimed at these less-represented pathologies could help affected patients or family members who turn to this social network in search of support or other purposes.

On the other hand, both the personal experiences of relatives of patients and family associations that claim to support families, and raising awareness and encouraging broader research of neurological diseases, were the most commented on topics found in our study in the Spanish and English tweets. We observed that sadness and fear were the most clearly associated sentiments, although anger and surprise are reported in our work. Compelling evidence has claimed that the impact of neurological disorders on the relatives of affected patients is a largely underestimated issue and needs more political and public awareness [33,34]. Thus, understandably, Twitter can be an ideal platform to turn to in order to express your needs and seek social and community support. Different studies that have been conducted on Twitter have obtained similar results in this sense. Yoon et al. [35] collected tweets mentioning dementia/Alzheimer's family caregiving-related terms for a period of one year before and during the coronavirus disease 19 (COVID-19) pandemic. They observed that the number of tweets mentioning family caregiver emotional distress, such as depression, helplessness and loneliness, and those detailing coping strategies and resilience topics for dementia caregiving increased during the COVID-19 pandemic. Klein et al. [29] used Twitter as an important tool with which to identify the relatives of patients with dementia, proposing its use to not only explore family caregivers' experiences, but to also directly target interventions to these users. Additionally, previous works

have also found a significant number of tweets perpetuating social stigmas and showing a general lack of knowledge of neurological diseases [27,36,37], which might explain the tweets aimed at raising public awareness on this social media. Likewise, users have also asked for further research in the field of neurological diseases. We observed that charity fundraising organizations for the study and research of neurological disorders also represented a significant number of English tweets. Twitter is a valuable platform for promoting the research of neurological disorders, enhancing the visibility of and recruitment for primary investigations, facilitating international connections and networks, allowing the rapid communication of their results, reaching a broader audience, et cetera [13]. Indeed, Hrincu et al. [28] found that academics and researchers are the largest user group discussing tweets related to neurological disorders like dementia. However, our results may indicate that Twitter users perceive the current research to be insufficient, or that the research collected in this field does not have sufficient dissemination, demonstrating the need to improve these aspects through this platform.

Simultaneously, the need/use of certain medications for the treatment of neurological diseases (particularly for the treatment of Parkinson's disease) and the impact of neurological diseases on daily life match the second and third most commented on topics in the Spanish tweets, whereas in the English tweets, these correspond to the exploration of new treatments and the pathophysiological bases of neurological disorders and those related to the research of neurological disorders, respectively. In the case of the use of medications available for Parkinson's, we observed that joy was commonly manifested in our studied tweets. Prior works have demonstrated the usefulness of Twitter for studying the perceptions, beliefs and sentiments related to pharmacotherapy in distinct pathologies [38]. Recent surveys conducted on physicians and patients with Parkinson's disease have shown that both groups have reported moderate satisfaction with the available treatments [39]. Thus, it is likely that something similar could be observed on Twitter, although the types of users were not identified in this study. The impact of neurological disorders on daily life and the way in which they can be faced was also a matter of discussion in the Spanish tweets, and this has been observed for other diseases in which patients and other users have shared and discussed their daily experiences in dealing with their conditions [40,41]. On the other hand, we observed that in the English tweets, there was a growing interest in novel and potential treatments for neurological disorders, as well as pathophysiological mechanisms and research and, despite our sentiment analysis showing a significant number of tweets displaying joy, the predominant emotion by far was fear. Unraveling the pathological basis of neurological disorders represents an important challenge to be faced by both healthcare professionals and researchers. Nowadays, it is broadly accepted that each neurological disorder has some singularities in the mechanisms and pathological clues related to its development; however, other biological hallmarks, such as neuroinflammation, gut dysbiosis or oxidative stress, are shared by virtually all chronic and neurological disorders [42,43]. As different investigations bring up novel mechanisms implicated in the development of neurological disorders, it is understandable that Twitter users would react with fear or negative emotions. Something similar could be happening with the different treatments currently being explored, in which there are plenty of pharmacological (i.e., natural compounds like cannabis or epigallocatechin, modulators of gut microbiota) and non-pharmacological (lifestyle factors like exercise and music, stem cell therapies, cognitive stimulation therapy) interventions that are being investigated in this field [44–47]. As mentioned above, there are a significant number of researchers, academics and institutions on Twitter showing their latest results, and although some users receive their studies with joy and positive emotions, a significant number of users are not at all confident. This may be because there is an excess of misinformation both on the Internet and on social networks, such as Twitter, that can make users distrust these novel treatments. In the same manner, prior works have also reported that messages shared on Twitter are not always transmitted effectively, and there are also a significant number of non-academics who are interested enough in the research to tweet articles, claiming the lack of academics' ability to write tweets that

are understandable to a non-specialist audience [48]. In this sense, our results show the need to change the transmission methods for novel and promising therapeutic approaches to neurological disorders, and for a greater involvement of scientists and clinicians on this platform.

This research has certain restrictions. First off, the collected tweets might not accurately represent society as a whole. Second, it is possible that tweets using contractions or slang to describe neurological disorders could have been unnoticed by this work. Finally, a certain degree of subjectivity is inherent in the process of topic modeling, as it is needed to group different words around a single theme or idea. In line with this, our results could have been influenced by the type of methodology that we used. BERTopic and other contextual topic models represent a transformative shift in natural language processing, leveraging contextual embeddings, such as BERT's bidirectional understanding, to capture subtle word meanings in sentences and documents [49,50]. These models offer distinct advantages over the application of LDA, the limitation of which lies in its treatment of words as independent terms, hindering the detection of the contextual relationships between them. This deficiency particularly impacts the identification of figures of speech and polysemy. However, we decided to choose LDA to detect topics due to its simplicity and widespread application, as evidenced in the existing work on Twitter [51–53]. In this scenario, the weight schema employed was based on a term's frequency of occurrence in each document. Given the nature of the text, tweets usually have a very short length in which to discuss neurological disorders, diminishing the vocabulary size. Truica et al. demonstrated, using the same implementation of LDA that we adopted, that this approach has proven to be the most effective choice in situations with limited vocabulary sizes, particularly when compared to the TF-IDF method [54]. However, this is not the only method with which to analyze Twitter data based on keywords, as the literature has recognized different options that have been considered in other works [55–57]. Likewise, there are also available approaches to clustering documents using embeddings or topic labels [58,59]. However, employing embeddings alongside a non-specialized text clustering algorithm introduces complexity and extends the computational time. The adoption of these techniques typically involves four stages: preprocessing, feature vectorization, clustering methods and the detection of topics within the clusters. In contrast, leveraging LDA necessitates only two steps: preprocessing and applying the LDA model. Also, despite our use of the distilROBERTa database, topic modeling techniques like proper LDA could be used to deepen the sentiment analysis, as has been performed in other works [60]. The use of information diffusion models, such as the MABED (Modeling and Analysis of Burst Event Dynamics) or Peaky Topics algorithm, to study the temporal evolution of topics has also been considered in other works [61]. Finally, despite not being considered herein, there has also been research on the analysis of emotions expressed during specific events on Twitter and on recent approaches proposing new ensemble architectures [62]. Ensemble architectures often combine multiple models and techniques to improve the accuracy and robustness of sentiment analysis for capturing the emotional nuances of Twitter data during an event.

Overall, for future directions of research, more innovative density-based models, such as the one reported in this study, should be highlighted [63]. Sentiment analysis using topic-document embeddings [64] could enhance clustering. Additionally, it has been demonstrated that the use of document vectorization techniques before clustering can improve the final result. Lastly, to enhance emotion detection, some methods incorporating topic modeling into the classification model have been developed, as reported in [56].

## 5. Conclusions

Our study shows that not only is Twitter an important and active platform implicated in the dissemination and revindication of neurological disorders, but also that the number of tweets discussing the different disorders is quite inequitable, with little representation of important neurological disorders like spinal cord injury. Likewise, a greater intervention

and the better dissemination of information by different figures and professionals on this social media platform could help to convey an understanding of State-of-the-Art information about neurological diseases among the general public, thereby reducing sentiments of fear, sadness and anger, and favoring moderate optimism toward the present and future of neurological diseases.

**Author Contributions:** Conceptualization, J.D.-E., O.F.-M. and M.A.A.-M.; methodology, J.D.-E., O.F.-M., C.G.-M., M.A.O., M.A.-M. and M.A.A.-M.; software, M.M., A.V. and F.J.L.-A.; validation, J.D.-E., M.A.O., M.A.-M. and M.A.A.-M.; formal analysis, J.D.-E., M.M., A.V. and F.J.L.-A.; investigation, J.D.-E., O.F.-M. and C.G.-M.; resources, M.A.O., M.A.-M. and M.A.A.-M.; data curation, M.M.; writing—original draft preparation, J.D.-E., O.F.-M. and C.G.-M.; writing—review and editing, M.A.O., M.A.-M. and M.A.A.-M.; visualization, J.D.-E., O.F.-M., C.G.-M. and M.A.O.; supervision, M.A.-M. and M.A.A.-M.; project administration, M.A.O., M.A.-M. and M.A.A.-M.; funding acquisition, M.A.O., M.A.-M. and M.A.A.-M. All authors have read and agreed to the published version of the manuscript.

**Funding:** This study was funded by the Instituto de Salud Carlos III (ISCIII) through project "PI22/00653", and was cofunded by the European Union, as well as by P2022/BMD-7321 (Comunidad de Madrid), ProACapital, Halekulani S.L. and MJR.

**Institutional Review Board Statement:** Not applicable.

**Informed Consent Statement:** Not applicable.

**Data Availability Statement:** The data used to support the findings of the present study are available from the corresponding authors upon request.

**Conflicts of Interest:** The authors declare no conflicts of interest. Javier Domingo-Espiñeira is the Business Unit Director, CNS and WHC, at Adamed Pharma. However, Adamed Pharma did not participate in this research.

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
