# Peer review of "Navigating the Digital Neurolandscape: Analyzing the Social Perception of and Sentiments Regarding Neurological Disorders through Topic Modeling and Unsupervised Research Using Twitter"

_information, doi:10.3390/info15030152_

Round 1
Reviewer 1 Report
Comments and Suggestions for Authors
The article analyzes social media to determine the discourse around neural disorders. The authors use both English and Spanish tweets. The analysis is three-fold, focusing on the temporal evolution of the discourse, topic modeling, and sentiment analysis.
The work is missing a related work on the following topics:
- There is no discussion on the topic models like BERTopic or contextual topic models [1]
- There is no discussion topic modeling performance when taking into account different weighting schemas [2]
- Other methods for grouping documents based on keywords are discussed in the current literature [3], [4], [5].
- There is a lot of literature on how to cluster documents using document embeddings [6] or topic labels [7].
- On the task of sentiment analysis, the current literature discusses methods to enhance polarity detection using topic modeling [8], [9]
- Regarding the temporal evolution of topics, there is a huge body of work on information diffusion [10], for example, MABED or Peaky Topics as algorithms
- There is a large body of work that analyzes the sentiment of Twitter events [11], and some current methods propose novel ensemble architectures [12]
The experimental section falls short:
- Why use LDA? Why not LSI or contextual topic modeling algorithms? Why not any of the algorithms from the provided Google Scholar links? There are many approaches for grouping textual data, so why not any of these?
- How has the text vectorized? LDA works with several document vectorizations like TF or TFIDF, see for example [2].
- There is no quantitative or qualitative analysis of the topics, for example, ARI or Coherence
- There is no evaluation regarding the sentiment analysis.
- Why choose the "Emotion English DistilRoBERTa-base" and the "Beto emotion analysis"? Did you perform any benchmarks to determine the correct models for your datasets? Were these models chosen out of convenience?
For reproducibility purposes, the code and dataset should be made publicly available.
The work is missing future direction.
The article needs proofreading.
Some sentences make no sense, e.g., "The silhouette coefficient was 102 the CVI selected, it ranges between -1 and 1, with higher values indicating better clustering performance."
[1] https://scholar.google.com/scholar?hl=en&as_sdt=0%2C5&q=%22topic+modeling%22+%2B+%22contextual+cues%22&btnG=
[2] https://scholar.google.com/scholar?hl=en&as_sdt=0%2C5&q=%22topic+modeling%22+%2B+%22weighting+schemas%22&btnG=
[3] https://scholar.google.com/scholar?hl=en&as_sdt=0%2C5&q=DenLAC&btnG=
[4] https://scholar.google.com/scholar?hl=en&as_sdt=0%2C5&q=evaluation+%2B+DBSCAN+%2B+similarity+join&btnG=
[5] https://scholar.google.com/scholar?hl=en&as_sdt=0%2C5&q=%22community+detection%22+%2B+%22document+clustering%22&btnG=
[6] https://scholar.google.com/scholar?hl=en&as_sdt=0%2C5&q=%22clustering+documents%22+%2B+%22document+to+vector+model%22&btnG=
[7] https://scholar.google.com/scholar?hl=en&as_sdt=0%2C5&q=%22topic+labeling%22+%2B+%22term+recognition%22&btnG=
[8] https://scholar.google.com/scholar?hl=en&as_sdt=0%2C5&q=%22sentiment+analysis%22+%2B+topic+%2B+%22document+embeddings%22&btnG=
[9] https://scholar.google.com/scholar?hl=en&as_sdt=0%2C5&q=document-level+sentiment+analysis+%2B+contextual+cues&btnG=
[10] https://scholar.google.com/scholar?hl=en&as_sdt=0%2C5&q=event+detection+on+twitter&btnG=&oq=event+detection+on+tittwe
[11] https://scholar.google.com/scholar?hl=en&as_sdt=0%2C5&q=%22sentiment+analysis+of+events%22+%2B+social+media+%2B+information+diffusion&btnG=
[12] https://scholar.google.com/scholar?hl=en&as_sdt=0%2C5&q=event+detection+%2B+sentiment+analysis+%2B+ensemble+architecture&btnG=
Comments on the Quality of English Language
The article needs proofreading
Author Response
REVIEWER 1
The article analyzes social media to determine the discourse around neural disorders. The authors use both English and Spanish tweets. The analysis is three-fold, focusing on the temporal evolution of the discourse, topic modeling, and sentiment analysis.
The work is missing a related work on the following topics:
- There is no discussion on the topic models like BERTopic or contextual topic models [1]
BERTopic and contextual models offer distinct advantages over the application of the LDA application. LDA's limitation lies in its treatment of words as independent terms, hindering the detection of contextual relationships between them. This deficiency particularly impacts the identification of figures of speech and polysemy. However, we decided to choose LDA to detect the topics due to its simplicity and widespread application, as evidenced in existing work on Twitter [1,2,3,4]. Our research primarily focuses on applying a well-documented technique within a novel database, aiming to extract meaningful insights from the data.
- There is no discussion topic modeling performance when taking into account different weighting schemas [2].
In this scenario, the weight schema employed is based on the term's frequency of occurrence in each document. Given the nature of the text, tweets usually have a very short length and discuss about neural disorders, diminishing the vocabulary size. Truica et al demonstrated using the same implementation of LDA that we have adopted, that this approach proves to be the most effective choice in situations with limited vocabulary sizes, particularly when compared to the TF-IDF method [5].
- Other methods for grouping documents based on keywords are discussed in the current literature [3], [4], [5].
Density-based models have gained interest in recent years, being the detection of outliers one of their main advantages Notably, some density-based models are now being applied in Twitter databases [6, 7]. Nevertheless, this type of clustering requires more computation effort compared to LDA. Density-based methods necessitate a prior feature vectorization of the documents. Additionally, to extract valuable insights post-clustering, an additional step is needed to identify the most crucial words to enhance interpretability. In contrast, utilizing LDA obviates the need for these steps, as they are inherently embedded within the model.
- There is a lot of literature on how to cluster documents using document embeddings [6] or topic labels [7].
The use of document embeddings are growing in interest, demonstrating superior results in certain scenarios [8]. However, employing embeddings alongside a non-specialized text clustering algorithm introduces complexity and extends computational time. The adoption of these techniques typically involves four stages: preprocessing, feature vectorization, clustering methods, and the detection of topics within clusters. In contrast, leveraging LDA necessitates only two steps: preprocessing and applying the LDA model. Notably, LDA is a well-established model with proven efficacy, currently employed in Twitter databases [1,2,3,4]. The main focus of this project is the application of well-established techniques to extract insights in medical themes. Thus, we determined that LDA represents the most suitable approach, offering efficiency and reliability for our objectives.
- On the task of sentiment analysis, the current literature discusses methods to enhance polarity detection using topic modeling [8], [9]
We highly value the suggestions you provided, and indeed, the incorporation of topics into document embeddings presents a fascinating approach. This integration has the potential to enhance sentiment analysis performance, mitigating limitations associated with figures of speech and irony. However, following the indication of the psychiatrists our primary focus in this part is detecting Ekman's six emotions plus a neutral class. Nevertheless, we are keen to explore the integration of topics into document embeddings in future projects, recognizing its potential for further improvement in sentiment analysis outcomes.
- Regarding the temporal evolution of topics, there is a huge body of work on information diffusion [10], for example, MABED or Peaky Topics as algorithms.
We appreciate your feedback, and for future projects, we plan to incorporate more advanced techniques to analyze clusters over time. In the current case, we opted for LDA due to its general applicability and for practical reasons. However, we are open to exploring novel algorithms and methodologies to enhance our temporal cluster analysis in upcoming projects.
- There is a large body of work that analyzes the sentiment of Twitter events [11], and some current methods propose novel ensemble architectures [12]
We appreciate your feedback, and moving forward, we intend to explore more innovative techniques for sentiment analysis. However, for this specific project, the comprehensive analysis of emotions, as determined by physicians, led us to prioritize the detection of Ekman's six emotions along with a neutral category. This approach aligns with the project's requirements for a thorough understanding of the emotional spectrum.
The experimental section falls short:
- Why use LDA? Why not LSI or contextual topic modeling algorithms? Why not any of the algorithms from the provided Google Scholar links? There are many approaches for grouping textual data, so why not any of these?
LDA exhibits several advantages when compared to other models, such as LDI. LDA, being a generative model, is particularly well-suited for handling large-scale corpora. Its superior performance in topic determination is noteworthy, as the topics generated are generally more interpretable for humans [9]. Among the various models available, we opted for LDA in this study due to its simplicity, efficiency, and widespread usage in contemporary research [1,2,3,4].
- How has the text vectorized? LDA works with several document vectorizations like TF or TFIDF, see for example [2].
In the version of LDA that we implemented, we generated a Document-Term Matrix (DTM). In this matrix, each document is represented as a row. To construct the vocabulary, we considered all the words present in the documents. The matrix entries indicate the frequency of each word within the respective document, providing a comprehensive representation of the corpus. We conclude that this vectorization was the best approach because Truica et al [5] demonstrated its superior performance with limited vocabulary sizes.
- There is no quantitative or qualitative analysis of the topics, for example, ARI or Coherence
We performed a quantitative analysis of topics by calculating the silhouette coefficient, a metric for coherence. This approach aided in determining the optimal number of topics and evaluating their quality. In order to select the number of topics for each database, we prioritized those achieving the highest coherence score. This method ensured a thorough assessment of the described topics, contributing to the overall quality of our analysis.
- There is no evaluation regarding the sentiment analysis.
The Emotion English DistilRoBERTa-base model demonstrated an accuracy of 66%, surpassing the random-chance baseline of 14% (1/7). This evaluation, conducted by the model's authors, encompassed six distinct English datasets [10]. On the other hand, the BETO model exhibited a macro F1-score of 55% in Spanish, as observed by its authors [11]. The notable performance of both models establishes their utility and effectiveness for analytical purposes.
- Why choose the "Emotion English DistilRoBERTa-base" and the "Beto emotion analysis"? Did you perform any benchmarks to determine the correct models for your datasets? Were these models chosen out of convenience?
Regarding the emotions detection, numerous models exist in the literature. However, it's worth highlighting that the models employed in this study are recognized as state-of-the-art for detecting Ekman's six basic emotions along with a neutral class. Capturing these emotions is crucial in our case, as per the insights from physicians who consider them to encompass all general feelings. The selected models have demonstrated superior performance in capturing these specified emotions. Furthermore, it is essential for our analysis that both the English and Spanish models consistently detect the same set of emotions to ensure the accuracy and reliability of our findings.
For reproducibility purposes, the code and dataset should be made publicly available.
We have no objection to showing the code to the reviewers or editors of the journal, and we are also willing to share it with any researcher who requests it in a justified manner. However, we prefer not to make it public to everyone because it has taken many hours of work, and in our opinion, if someone wants to use it, it should be within the framework of scientific collaboration.
The work is missing future direction.
We really welcome the important comment made by the reviewer. We have added this paragraph in the manuscript in order to clarify future direction
There are more innovative density-based models, such as the one reported in this study (Mitroi, M., Truică, C. O., Apostol, E. S., & Florea, A. M. (2020, September). Sentiment analysis using topic-document embeddings. In 2020 IEEE 16th international conference on intelligent computer communication and processing (ICCP) (pp. 75-82). IEEE.), that could enhance clustering. Additionally, it has been demonstrated that the use of document vectorization techniques before clustering can improve the final result. Lastly, to enhance emotion detection, some methods incorporating topic modeling into the classification model have been developed, as reported in this study: Rădulescu, I. M., Boicea, A., Truică, C. O., Apostol, E. S., Mocanu, M., & Rădulescu, F. (2021, June). DenLAC: Density Levels Aggregation Clustering–A Flexible Clustering Method–. In International Conference on Computational Science (pp. 316-329). Cham: Springer International Publishing.
The article needs proofreading.
Some sentences make no sense, e.g., "The silhouette coefficient was 102 the CVI selected, it ranges between -1 and 1, with higher values indicating better clustering performance."
[1] https://scholar.google.com/scholar?hl=en&as_sdt=0%2C5&q=%22topic+modeling%22+%2B+%22contextual+cues%22&btnG=
[2] https://scholar.google.com/scholar?hl=en&as_sdt=0%2C5&q=%22topic+modeling%22+%2B+%22weighting+schemas%22&btnG=
[3] https://scholar.google.com/scholar?hl=en&as_sdt=0%2C5&q=DenLAC&btnG=
[4] https://scholar.google.com/scholar?hl=en&as_sdt=0%2C5&q=evaluation+%2B+DBSCAN+%2B+similarity+join&btnG=
[5] https://scholar.google.com/scholar?hl=en&as_sdt=0%2C5&q=%22community+detection%22+%2B+%22document+clustering%22&btnG=
[6] https://scholar.google.com/scholar?hl=en&as_sdt=0%2C5&q=%22clustering+documents%22+%2B+%22document+to+vector+model%22&btnG=
[7] https://scholar.google.com/scholar?hl=en&as_sdt=0%2C5&q=%22topic+labeling%22+%2B+%22term+recognition%22&btnG=
[8] https://scholar.google.com/scholar?hl=en&as_sdt=0%2C5&q=%22sentiment+analysis%22+%2B+topic+%2B+%22document+embeddings%22&btnG=
[9] https://scholar.google.com/scholar?hl=en&as_sdt=0%2C5&q=document-level+sentiment+analysis+%2B+contextual+cues&btnG=
[10] https://scholar.google.com/scholar?hl=en&as_sdt=0%2C5&q=event+detection+on+twitter&btnG=&oq=event+detection+on+tittwe
[11] https://scholar.google.com/scholar?hl=en&as_sdt=0%2C5&q=%22sentiment+analysis+of+events%22+%2B+social+media+%2B+information+diffusion&btnG=
[12] https://scholar.google.com/scholar?hl=en&as_sdt=0%2C5&q=event+detection+%2B+sentiment+analysis+%2B+ensemble+architecture&btnG=
References
[1] Parker, M. A., Valdez, D., Rao, V. K., Eddens, K. S., & Agley, J. (2023). Results and Methodological Implications of the Digital Epidemiology of Prescription Drug References Among Twitter Users: Latent Dirichlet Allocation (LDA) Analyses. Journal of Medical Internet Research, 25, e48405.
[2] Uthirapathy, S. E., & Sandanam, D. (2023). Topic Modelling and Opinion Analysis On Climate Change Twitter Data Using LDA And BERT Model. Procedia Computer Science, 218, 908-917.
[3] Cao, Z., Hu, S., & Tao, Z. (2023, January). Analyze comments on Twitter about extreme weather based on the Latent Dirichlet Allocation (LDA) approach. In Third International Conference on Intelligent Computing and Human-Computer Interaction (ICHCI 2022) (Vol. 12509, pp. 51-56). SPIE.
[4] Pamula, A., Gontar, Z., Gontar, B., & Fesenko, T. (2023). Latent Dirichlet Allocation in Public Procurement Documents Analysis for Determining Energy Efficiency Issues in Construction Works at Polish Universities. Energies, 16(12), 4596.
[5] Truica, C. O., Radulescu, F., & Boicea, A. (2016, September). Comparing different term weighting schemas for topic modeling. In 2016 18th international symposium on symbolic and numeric algorithms for scientific computing (SYNASC) (pp. 307-310). IEEE.
[6] Indah, R. N. G., Novita, R., Kharisma, O. B., Vebrianto, R., Sanjaya, S., Andriani, T., ... & Rahim, R. (2019, November). DBSCAN algorithm: twitter text clustering of trend topic pilkada pekanbaru. In Journal of physics: conference series (Vol. 1363, No. 1, p. 012001). IOP Publishing.
[7] Campello, R. J., Kröger, P., Sander, J., & Zimek, A. (2020). Density‐based clustering. Wiley Interdisciplinary Reviews: Data Mining and Knowledge Discovery, 10(2), e1343.
[8] Curiskis, S. A., Drake, B., Osborn, T. R., & Kennedy, P. J. (2020). An evaluation of document clustering and topic modelling in two online social networks: Twitter and Reddit. Information Processing & Management, 57(2), 102034.
[9] Liu, Z., Li, M., Liu, Y., & Ponraj, M. (2011, July). Performance evaluation of Latent Dirichlet Allocation in text mining. In 2011 Eighth International Conference on Fuzzy Systems and Knowledge Discovery (FSKD) (Vol. 4, pp. 2695-2698). IEEE.
[10] Jochen Hartmann, "Emotion English DistilRoBERTa-base". https://huggingface.co/j-hartmann/emotion-english-distilroberta-base/, 2022.
[11] Cañete, J., Chaperon, G., Fuentes, R., Ho, J. H., Kang, H., & Pérez, J. (2023). Spanish pre-trained bert model and evaluation data. arXiv preprint arXiv:2308.02976.
Reviewer 2 Report
Comments and Suggestions for Authors
The paper proposes one interesting idea but requires some more effort to be publishable.
1. The use of social networks is a good idea for this analysis. But due to the current movement on Twitter (now X) the study would be in better shape if it spanned multiple social networks.
2. The used of text mining and sentiment analysis methods is quite old, consider trying the novel ones.
3. Is the study resilient to multi-lingual and figure of speech? Since it relies on sentiment analysis on a social network these two tend to be the major limitations.
Comments on the Quality of English LanguageNo comments.
Author Response
REVIEWER 2
The paper proposes one interesting idea but requires some more effort to be publishable.
- The use of social networks is a good idea for this analysis. But due to the current movement on Twitter (now X) the study would be in better shape if it spanned multiple social networks.
Thank you very much for your comment. We decided to choose Twitter for several reasons: 1. It is among the most widely used social media platforms globally; 2. Posts primarily consist of text, whereas other social media platforms like TikTok or Instagram are more visual since posts are predominantly photos and videos; 3. It is the most widely used social media platform in the West for the discussion of various topics; 4. The majority of accounts are open; 5. The information can be purchased from the company. Other social media platforms such as Facebook or Instagram do not provide this option.
It would be interesting to replicate this study on an Eastern-origin social media platform that has similar characteristics to Twitter. Finally, the changes on Twitter occurred after the data collection, so we believe they do not affect the study.
- The used of text mining and sentiment analysis methods is quite old, consider trying the novel ones.
We appreciate your suggestion to explore novel approaches for the emotion detection and the topic modelling.
There are indeed various models for emotion detection in the literature but it's noteworthy that the used models are considered being the state of the art for detecting Ekman's six basic emotions plus a neutral class. In our analysis, we have specifically employed the Spanish model launched in 2021 and the English model launched in 2022. These models have demonstrated superior performance in capturing the previously specified emotions. Furthermore, it is essential for our analysis that both the English and Spanish models consistently detect the same set of emotions to ensure the accuracy and reliability of our findings.
Regarding the topic modeling, LDA is a well-established model with proven efficacy, currently employed in Twitter databases [1,2,3,4].Moreover, its versatility makes it a valuable tool for extracting meaningful information from unstructured text. The main focus of this project is the application of a well-established model in novel databases and thematic analysis. Therefore, we determined that LDA represents the most suitable approach, offering efficiency and reliability for our objectives.
- Is the study resilient to multi-lingual and figure of speech? Since it relies on sentiment analysis on a social network these two tend to be the major limitations.
To tackle the multilingual aspect, we categorized tweets into English and Spanish, using Google Translator to translate tweets from other languages into English. This approach effectively addresses the multilingual limitation.
Concerning figures of speech and irony, LDA topic modeling may face some limitations challenges . However, the sentiment analysis models employed in our study showcase a level of adaptability, enabling them to recognize specific figures of speech and instances of irony [5]. Hence, while acknowledging these challenges, our study adopts practical strategies to navigate the complexities of multilingual content and figurative language.
References:
[1] Parker, M. A., Valdez, D., Rao, V. K., Eddens, K. S., & Agley, J. (2023). Results and Methodological Implications of the Digital Epidemiology of Prescription Drug References Among Twitter Users: Latent Dirichlet Allocation (LDA) Analyses. Journal of Medical Internet Research, 25, e48405.
[2] Uthirapathy, S. E., & Sandanam, D. (2023). Topic Modelling and Opinion Analysis On Climate Change Twitter Data Using LDA And BERT Model. Procedia Computer Science, 218, 908-917.
[3] Cao, Z., Hu, S., & Tao, Z. (2023, January). Analyze comments on Twitter about extreme weather based on the Latent Dirichlet Allocation (LDA) approach. In Third International Conference on Intelligent Computing and Human-Computer Interaction (ICHCI 2022) (Vol. 12509, pp. 51-56). SPIE.
[4] Pamula, A., Gontar, Z., Gontar, B., & Fesenko, T. (2023). Latent Dirichlet Allocation in Public Procurement Documents Analysis for Determining Energy Efficiency Issues in Construction Works at Polish Universities. Energies, 16(12), 4596.
[5] Agrawal, A., Jha, A. K., Jaiswal, A., & Kumar, V. (2020, August). Irony detection using transformers. In 2020 International Conference on Computing and Data Science (CDS) (pp. 165-168). IEEE.
Round 2
Reviewer 1 Report
Comments and Suggestions for Authors
The quality of the manuscript has greatly improved over the previous version. I think it can be accepted in its current form after minor edits.
Comments: reference 64 is not correctly formatted and it should be fixed.
Comments on the Quality of English LanguageThe English is ok.